# A Transcriptomic Analysis of Head and Neck Squamous Cell Carcinomas for Prognostic Indications

**DOI:** 10.3390/jpm11080782

**Published:** 2021-08-11

**Authors:** Li-Hsing Chi, Alexander T. H. Wu, Michael Hsiao, Yu-Chuan (Jack) Li

**Affiliations:** 1The Ph.D. Program for Translational Medicine, College of Medical Science and Technology, Taipei Medical University and Academia Sinica, Taipei 11031, Taiwan; d622101005@tmu.edu.tw (L.-H.C.); chaw1211@tmu.edu.tw (A.T.H.W.); 2Division of Oral and Maxillofacial Surgery, Department of Dentistry, Wan Fang Hospital, Taipei Medical University, Taipei 11600, Taiwan; 3Division of Oral and Maxillofacial Surgery, Department of Dentistry, Taipei Medical University Hospital, Taipei Medical University, Taipei 11031, Taiwan; 4Genomics Research Center, Academia Sinica, Taipei 115024, Taiwan; 5Department of Biochemistry, College of Medicine, Kaohsiung Medical University, Kaohsiung 807378, Taiwan; 6Graduate Institute of Biomedical Informatics, College of Medical Science and Technology, Taipei Medical University, No.172-1, Sec. 2, Keelung Rd., Taipei 106339, Taiwan

**Keywords:** head and neck squamous cell carcinoma (HNSCC), the Cancer Genome Atlas (TCGA), transcriptomic analysis, survival analysis, optimal cutoff, effect size, calcium/calmodulin dependent protein kinase II inhibitor 1 (CAMK2N1), calmodulin like 5 (CALML5), Fc fragment of IgG binding protein (FCGBP), mindfulness meditation

## Abstract

Survival analysis of the Cancer Genome Atlas (TCGA) dataset is a well-known method for discovering gene expression-based prognostic biomarkers of head and neck squamous cell carcinoma (HNSCC). A cutoff point is usually used in survival analysis for patient dichotomization when using continuous gene expression values. There is some optimization software for cutoff determination. However, the software’s predetermined cutoffs are usually set at the medians or quantiles of gene expression values. There are also few clinicopathological features available in pre-processed datasets. We applied an in-house workflow, including data retrieving and pre-processing, feature selection, sliding-window cutoff selection, Kaplan–Meier survival analysis, and Cox proportional hazard modeling for biomarker discovery. In our approach for the TCGA HNSCC cohort, we scanned human protein-coding genes to find optimal cutoff values. After adjustments with confounders, clinical tumor stage and surgical margin involvement were found to be independent risk factors for prognosis. According to the results tables that show hazard ratios with Bonferroni-adjusted *p* values under the optimal cutoff, three biomarker candidates, CAMK2N1, CALML5, and FCGBP, are significantly associated with overall survival. We validated this discovery by using the another independent HNSCC dataset (GSE65858). Thus, we suggest that transcriptomic analysis could help with biomarker discovery. Moreover, the robustness of the biomarkers we identified should be ensured through several additional tests with independent datasets.

## 1. Introduction

Head and neck squamous cell carcinoma (HNSCC), including that of oral, oropharyngeal, and hypopharyngeal origins, is the fourth leading cause of cancer-related death for males in Taiwan [1]. The age-standardized incidence rate of HNSCC in males is 42.43 per 100,000 persons [2]. The treatment strategies of HNSCC are surgery alone, systemic therapy with concurrent radiation therapy (systemic therapy/RT), and surgery with adjuvant systemic therapy/RT (according to National Comprehensive Cancer Network, NCCN, Clinical Practice Guidelines for HNSCC, Version 2.2020) [3]. Despite the improvements in those interventions, the survival of HNSCC has improved only marginally over the past decade worldwide [4]. The critical advancements of targeted therapy and immuno-oncology should benefit from emerging prognostic biomarkers guiding modern systemic therapies.

Cumulative knowledge shows that some biomarkers have prognostic significance in HNSCC. For example, node-negative HNSCC patients with p53 overexpression were found to have lower survival [5]. Overexpression of hypoxia-inducible factor (HIF)-1 alpha [6] or Ki-67 [7] was found to be correlated with poor response to radiotherapy of HNSCC. Both epidermal growth factor receptor (EGFR) [8,9] and matrix metalloproteinase (MMP) [10] were found to be overexpressed to promote the invasion and metastasis of HNSCC. From 2000 to 2006, the first anti-EGFR antibody-drug (cetuximab) was developed and combined with radiotherapy, known as bio-RT, to increase survival with unresectable locoregionally advanced disease [11]. The systemic therapy of cetuximab plus platinum-fluorouracil chemotherapy (EXTREME regimen) improves overall survival when given as a first-line treatment in patients with recurrent or metastatic HNSCC [12,13]. It was approved by the US Food and Drug Administration (FDA) in 2008. The bio-RT could be proceeded with docetaxel, cisplatin, and 5-fluorouracil (Tax-PF) induction chemotherapy to overcome radio-resistance of HNSCC [14].

However, Rampias and his colleagues [15] suggested that Harvey rat sarcoma viral oncoprotein (HRAS) mutations could mediate cetuximab resistance in systemic therapy of HNSCC via the EGFR/rat sarcoma (RAS)/extracellular signal-regulated kinases (ERK) signaling pathway. After that, the EGFR tyrosine kinase inhibitor (TKI) was introduced to help cetuximab in 2018. Anti-tumor activity was observed in a phase 1 trial for HNSCC patients using cetuximab and afatinib, a TKI of EGFR, human epidermal growth factor receptor (HER)2, and HER4 [16]. Other EGFR TKIs, such as gefitinib, erlotinib, and osimertinib, were also developed to treat advanced HNSCC. Although 90% of HNSCCs overexpress EGFR, cetuximab only has a 10–20% response rate in those patients. As of 2019, cetuximab was still the only drug of choice with proven efficacy for selected HNSCC patients [17].

In the immuno-oncology era, the immune-checkpoint inhibitor (ICI) was introduced in 2014 for treating HNSCC [18,19]. The ICI works on immune checkpoint molecules, including programmed death 1 (PD-1), cytotoxic T lymphocyte antigen 4 (CTLA-4), T-cell immunoglobulin mucin protein 3 (TIM-3), lymphocyte activation gene 3 (LAG-3), T cell immunoglobin and immunoreceptor tyrosine-based inhibitory motif (TIGIT), glucocorticoid-induced tumor necrosis factor receptor (GITR), and V-domain Ig suppressor of T-cell activation (VISTA) [20]. The US FDA has approved the anti-PD-1 agents (e.g., pembrolizumab and nivolumab) as monotherapies for platinum-treated patients with recurrent or metastatic HNSCC [21]. According to the phase 3 KEYNOTE-048 study, PD-L1 is a validated biomarker used as clinical guidance for candidate selection of pembrolizumab [22,23]. However, due to the complexity of immune-tumor interactions, ICI has 20% response rate in programmed death ligand 1 (PD-L1)-expressing patients (over 50% in immunohistochemistry (IHC) staining of HNSCC) [19,23].

According to our previous proteomic study from 2010 to 2017, thymosin beta-4 X-linked (TMSB4X) is related to tumor growth and the metastasis of HNSCC [24]. It was also reported by the subsequent investigations that TMSB4X contributes to tumor aggressiveness through epithelial-mesenchymal-transition (EMT) in pancreatic [25], gastric [26], colorectal [27], lung [28], ovarian [29], and melanoma [30] cancers. Thus, it might be suggested that TMSB4X is a candidate for tumor type-agnostic therapy [31], as a common biomarker of several types of cancer.

The Cancer Genome Atlas (TCGA) has clinical and genomic data of HNSCC (528 participants), which were standardized and are available at a unified data portal, Genomic Data Commons (GDC) of the the National Cancer Institute (NCI). The advantages of applying the TCGA data for cancer biomarker identification include:To the best of our knowledge, the TCGA database is the largest collection (in terms of both cancer types and cohort size, especially in HNSCC) of comprehensive genomics with survival data available in the field of cancer research. The whole-genome sequencing data were harmonized across all genome data analysis centers. Many databases adopt the essential demographic data from TCGA, since it has comprehensive physical and social features of patients, such as exposure to alcohol, asbestos, radioactive radon, tobacco smoking, and cigarettes.TCGA has a remarkable advantage for computational and life scientists who study cancer, since useful web-based tools and APIs are ready to analyze and visualize TCGA data. It might be getting help soon from the research community for trouble-shooting purposes.Many achievements in diagnosis, treatment, and prevention that relied on the TCGA data have already been published and keep increasing in number [32].


Usually, researchers develop an in-house workflow of gene expression analysis of TCGA data to find HNSCC biomarkers. It would be helpful to show that alterations in gene expression correlate with phenotypes of HNSCC. Some researchers [33,34,35,36,37,38,39] tried to find differentially expressed genes (DEGs) of the HNSCC samples at both genotypic and phenotypic levels (without survival information) for biomarker discovery. Gene expression data were downloaded from the TCGA or Gene Expression Omnibus (GEO) databases (e.g., GSE117973 [39]; HIPO-HNC cohort has *n* = 87). They used the Database for Annotation, Visualization, and Integrated Discovery (DAVID, available at https://david.ncifcrf.gov/, accessed 15 November 2019) to obtain information for Gene Ontology (GO), including biological processes, cellular components, and molecular functions. Kyoto Encyclopedia of Genes and Genomes (KEGG) pathway analysis was also used to annotate the potential functions of their biomarker candidates. The pathway enrichment analysis of DEGs was also performed by DAVID, STRING (available at https://string-db.org, accessed 15 November 2019), and Cytoscape software [37,38]. Li [36] and his colleagues made an R package (GDCRNATool) for the implementation of those workflows for gene expression analyses of the TCGA. Xu and his colleagues [40] also identified their biomarkers via DEGs analysis. The significant impacts of genes on overall survival were evaluated with Kaplan–Meier survival curves with a log-rank test (*p* value < 0.01) and univariate Cox regression. They validated the candidate genes by using the web-based tools of Gene Expression Profiling Interactive Analysis—GEPIA—and Human Protein Atlas (HPA) databases. GEPIA was developed, using TCGA datasets, by Zefang Tang and his colleagues (version 1 [41], version 2 [42], and GEPIA2021 [43], available at http://gepia2021.cancer-pku.cn/, accessed 18 July 2021). HPA [44] applied immunohistochemistry (IHC) for the TCGA database (please see details in the Discussions section “Validation by Web-based Tools”). Finally, their biomarkers were verified by using the gene expression profile from the GEO and HNSCC cell lines and tissues.

Other investigators should gather genes of interest to specific cancer types. They should upload those genes manually onto web-based tools, such as SurvExpress [45] (available at http://bioinformatica.mty.itesm.mx:8080/Biomatec/SurvivaX.jsp, accessed 11 August 2021), and then analyze cohorts of interest (e.g., TCGA). After downloading the survival results, they could curate plots and tables carefully. It is not possible to scan the whole human protein-coding genome in this way. The web-based tools might set a cutoff at the median, 1/4 quantile, or 3/4 quantile for subsequent analyses. There are several visualization tools and R packages which deal with cutoff determination [46], such as Prognoscan [47], Cutoff Finder [48], Findcut [49], OptimalCutpoints [50], cutpointr (available at https://github.com/thie1e/cutpointr, accessed 20 November 2018), and cutoffR (available at https://cran.r-project.org/web/packages/cutoffR, accessed 27 June 2021). However, none of them could perform survival analysis in tandem with cutoff selection and whole-genome scanning.

In summary, identifying predictive biomarkers for selecting standard-of-care or advanced systemic therapy [50] in HNSCC is crucial. Our approach describes an in-house workflow implemented in R script, which runs on the Rstudio server. Its functions include data retrieving and pre-processing, feature selection, sliding-window cutoff selection, Kaplan–Meier survival analysis, Cox proportional hazard modeling, and biomarker discovery. The independent HNSCC dataset (GSE65858) [51] was used to validate this strategy. The workflow, shown in Figure 1, has scanned 20,500 human protein-coding genes of the TCGA HNSCC cohort to yield a model with biomarker estimates using gene-expression-based survival analysis.

## 2. Results

The TCGA HNSCC cohort was used for exploration of biomarker candidates. A total of 9416 Kaplan–Meier plots (under sliding-window cutoff selection) with associated Cox univariate and multivariate tables were generated by Cox modeling (Figure 1) and justified by the ranking of hazard ratios. In total, 967 out of 9416 genes were kept by criteria of FDR-adjusted Kaplan–Meier *p* values (<0.05) and hazard ratios (HR) derived from Cox’s model (Figure 2a,b, initial trial). In the next step, a stringent Bonferroni *p* value correction was used to yield 20 genes (Figure 2c,d).

CAMK2N1, CALML5, and FCGBP and 17 other genes (DKK1, STC2, PGK1, SURF4, USP10, NDFIP1, FOXA2, STIP1, DKC1, ZNF557, ZNF266, IL19, MYO1H, EVPLL, PNMA5, IQCN, and NPB) had significant FDR-adjusted *p* values (<0.0003) in the Kaplan–Meier estimates and appropriate hazard ratios (HRs) (>1.8 or <0.6) in Cox’s model (Figure 3; log${}_{10}\left(0.0003\right)=-3.5$). The volcano plot reveals that those top 20 genes (Bonferroni-adjusted *p* < 0.05) form the peaks. At the same time, Cox’s HRs separate them in regard to significant prognostic impact.

In our validation study using the GSE65858 cohort [51] (under median cutoffs), CAMK2N1, CALML5, and FCGBP (3 out of those 20 genes discovered in the TCGA cohort) kept ahead of the curve with their FDR-corrected *p* values (<0.05), and Cox’s HRs (>1.8 or <0.6) (Appendix A). However, the significance of the other 17 genes was insufficient compared to that of DUSP6, MSMB, and RBM11 (Figure 4). Conversely, there are 22 genes which have high hazard ratios (>1.8 or <0.6) in the GSE65858 cohort (Figure 4); their hazard ratios were between 0.6 and 1.5 in the study of TCGA HNSCC (Figure 3). Thus, there is a consensus between the TCGA and GSE65858 cohorts that CAMK2N1, CALML5, and FCGBP are significant candidates for HNSCC biomarkers.

Our top candidate is calcium/calmodulin dependent protein kinase II inhibitor 1 (CAMK2N1). The Kaplan–Meier curve reveals that 152 patients bearing higher expression levels of CAMK2N1 suffered from an only 35% 5-year OS rate. In comparison, the other 262 patients with lower expression levels had better prognoses (Bonferroni-adjusted *p* =0.002) (Figure 5a). Figure 5b’s cumulative *p* value plot shows that the 147 uncorrected *p* values (<0.05) were estimated by a serial cut from 144 to 290 persons for grouping the cohort in our cutoff finding procedure (cutofFinder_func.R; Figure 1, cutoff engine). The smallest *p* value (2.97 × 10^−7^), when cut at *n* = 262 (63.3% of total cohort 414, with the cutoff value of 0.027 for RNA-Seq by Expectation-Maximization—RSEM), was defined as an optimal *p* value. The plot in Figure 5b shows a “backlash” curve with the half of values below 1.0 × 10^−3^.

Conversely, the gene most associated with better survival was calmodulin like 5 (CALML5). In Figure 5c, a Kaplan–Meier curve reveals 200 patients bearing higher expression of CALML5 had a 60% 5-year OS survival rate (Bonferroni-adjusted *p* =0.039). The sliding-window cutoff-selection-generated cumulative *p* value plot is in Figure 5d. This plot reveals a “V” curve with the minimum at the middle portion. The 166 uncorrected *p* values were estimated by a serial cut from 125 to 290 for grouping the cohort. The smallest *p* value (5.87 × 10^−6^), when cut at *n* = 214 (51.7% of total cohort 414), was defined as an optimal *p* value with a cutoff value of −0.359 for RSEM.

The third candidate is Fc fragment of IgG binding protein (FCGBP). It was also correlated with better survival in both the TCGA and GSE65858 cohorts. In Figure 5e, a Kaplan–Meier curve reveals 282 patients bearing higher expression levels of FCGBP had a 60% 5-year OS survival rate (Bonferroni-adjusted *p* =0.008). The sliding-window cutoff-selection-generated cumulative *p* value plot is in Figure 5f. This plot has a “W-shaped” curve with the majority of values being far below 1.0 × 10^−3^. The 166 uncorrected *p* values were estimated by a serial cut from 125 to 290 for grouping the cohort. The smallest *p* value (1.21 × 10^−6^), when cut at *n* = 132 (31.9% of total cohort 414), was defined as an optimal *p* value with a cutoff value of −0.472 for RSEM.

After adjustments for confounders, CAMK2N1 overexpression became an independent prognostic factor (multivariate HR 2.007 (95% CI: 1.490–2.704, *p* < 0.001), Table 1). The clinical T stage (HR 1.982 (95% CI: 1.048–3.745, *p* = 0.035)) and surgical margin status (HR 1.631 (95% CI: 1.182–2.250, *p* = 0.003)) also have significant impacts on a patient’s survival. A patient being older than 65 could worsen survival (HR 1.391 (95% CI: 1.025–1.888, *p* = 0.034)). The M stage should be ignored in this cohort due to only 3 out of 414 patients having distant metastasis.

In summary, those three biomarker candidates, clinical T stage, and the presence of a surgical margin are independent prognostic factors in HNSCC. We also found those candidates have proper effect sizes—Cox’s HR >1.8 or <0.6; Table 2. Thus, the prognostic model with coefficients was established by the TCGA HNSCC cohort and validated by the GSE65858 cohort.

## 3. Discussion

### 3.1. The Three Biomarkers in Cancer

#### 3.1.1. The Protein/Pathology Atlas

Proteomics analysis in the Human Protein Atlas project (HPA) was based on 26,941 antibodies targeting 17,165 unique proteins. The HPA’s Pathology Atlas analyzed each protein in patients using immunohistochemistry (IHC) analysis based on tissue microarrays (TMAs) adopted from TCGA. Kaplan–Meier survival analyses were based on RNA-Seq expression levels of human genes in HNSCC tissue and the clinical outcome.

CAMK2N1 is on the list of unfavorable prognostic genes for HNSCC, and lung or liver cancer from the Human Protein Atlas (HPA) (pathology atlas [44] is available at https://www.proteinatlas.org/humanproteome/pathology/head+and+neck+cancer, Version: 20.0 updated: 19 November 2020, accessed 27 June 2021). CALML5 and FCGBP are on the list of favorable prognostic genes for HNSCC (available at https://www.proteinatlas.org/ENSG00000178372-CALML5/pathology; https://www.proteinatlas.org/ENSG00000275395-FCGBP/pathology, respectively). Furthermore, CALML5 was validated by the HNSCC cohort at mRNA and protein levels [44,52].

#### 3.1.2. Literature Review

We searched Embase/Pubmed to find the evidence of our three biomarker candidates in cancer research.

CAMK2N1 (calcium/calmodulin dependent protein kinase II inhibitor 1) is an endogenous inhibitor of calcium/calmodulin-dependent protein kinase II, CaMKII. CaMKII is a multi-functional kinase composed of four different chains: alpha, beta, gamma, and delta. CAMK2A encodes the alpha chain. Although the mRNA expression of CaMKII’s endogenous inhibitor CAMK2N1 inversely correlates with the severity of medullary thyroid carcinoma [53], both CAMK2N1 (HR=2.1) and CAMK2A (HR=1.6) were overexpressed in the TCGA HNSCC patientswith worse outcomes. Overexpression of CAMK2N1 is also unfavorable in head and neck cancer, as revealed by the Human Pathology Atlas (available at https://www.proteinatlas.org/ENSG00000162545-CAMK2N1/pathology/head+and+neck+cancer, accessed on 10 March 2021). Another web-based tool, KM plotter [54,55], shows that CAMK2N1 either worsens or improves survival in various cancer types.

Calmodulin-like 5 (CALML5) is overexpressed in differentiating keratinocytes [56]. In patients with HPV-associated HNSCC, hypermethylation of the CALML5 gene is associated with significantly reduced survival, with a hazard ratio of 7.01 (95% CI: 1.01–48.66) [57]. CALML5 expression could be a protective mechanism for patient survival. Furthermore, CALML5 was validated by the HNSCC cohort at mRNA and protein levels [44,52].

The Fc fragment of the IgG binding protein (FcγBP, FCGBP) is expressed in the normal thyroid and is down-regulated in papillary and follicular thyroid carcinomas [58,59]. Overexpression of FCGBP has hazard ratio of 0.306 (95% CI: 0.136–0.686) for gallbladder cancer [60].

In conclusion, the three prognostic genes underlined have been highlighted by published studies using the TCGA cohort, an in-house cohort, or in vitro and in vivo experiments.

### 3.2. Feature Selection for Survival Modeling

Besides ethnicity, age, gender, TNM stage, radiation therapy, chemotherapy, and targeted therapy, the comprehensive adversely prognostic features in HNSCC should also include tobacco exposure, EGFR amplification, human papillomavirus (HPV) status, positive/close surgical margin (<5 mm), extra-nodal extension (ENE), lymph-vascular space invasion (LVSI), perineural invasion (PNI), depth of invasion (DOI) (>5 mm), metastatic lymph node density (LND) [61], and worst pattern of invasion score 5 (WPOI-5), which is defined as tumor dispersion (1 mm apart between tumor satellites) or positive PNI/LVSI [62]. The features of DOI, LND, and tumor dispersion are not available in the TCGA dataset. The Brandwein–Gensler risk model (lymphocytic host response, WPOI-5, and PNI) [63,64] has been suggested for routine pathological examinations. In previous reports of HNSCC, the loco-regional failure was high when the initial frozen section had a positive/close surgical margin, and even the final margin revision revealed a negative effect [65]. According to Table 1, in our study, the positive surgical margin had to yield a hazard ratio greater than 1.6 to influence a patient’s OS. It is suggested by authors [66,67,68,69,70,71,72,73,74,75] that the reason for a positive/close surgical margin is possibly tumor aggressiveness or dispersion (WPOI-5) instead of iatrogenic actions in surgery. The surgical margin status was also suggested as an independent surrogate for tumor dispersion in the HNSCC study. Thus, we selected common clinicopathological features during our biomarker discovery, including gender, age, clinical T, clinical N, clinical M, surgical margin status, and tobacco exposure, to adjust confounders (details description at Materials and Methods section).

### 3.3. The Purpose of Sliding-Window Cutoff Selection

By trying to find an optimal cutpoint of that RNA expression data to maximize candidate mining coverage, this strategy could identify more but sometimes weak “biomarkers.” Thus, we had to try our best to handle the effect size with Cox’s modeling. Additionally, validation of those candidates was required by using other independent datasets.

Statistical significance (*p* value) is affected by sample size, error, effect size (substantive significance) [76,77], and cutoff. The effect size is the magnitude of the difference (e.g., hazard ratio) between the groups being compared. The effect size is independent of the sample size [76].

In a study with a large sample size, the difference can be noticed easily (i.e., *p* value < 0.05) due to decreased standard error [76]. However, a small effect size (non-zero) is often meaningless or implies substantive insignificance (e.g., hazard ratio between 0.8 and 1.2). Conversely, the effect size can be large but fail to gain statistical significance if the sample size is small. The following errors could also impact the *p* value:A random error, defined as the variability in data, is not considered a bias but rather occurs randomly across the entire study population and can distort the measurement process (e.g., RNA-Seq experiments). A larger sample size could reduce the random error.A systematic error is a bias, a selection biases, an information bias, or a confounder. It could deleteriously impact the statistical significance. A larger sample size could not affect the systematic error.

While statistical significance can inform the researcher of whether an effect exists, the *p* value will not directly tell one of the effect size. Thus, if there is no error in two study groups, and the sample sizes are the same (not small), the group which has a larger effect size will have a small *p* value [77]. If a skewed cutoff that splits between the two groups—for example, into 425 versus 75—statistical significance will be gained by increasing the effect size artificially.

There was a benefit in using the sliding-window cutoff method (the between 30% and 70% quantile) in Kaplan–Meier analysis of the TCGA HNSCC cohort at the beginning. We compared the results of cutoff at the optimal *p* value by sliding window or just at the median of gene expression. The numbers of genes (with unadjusted *p* values < 0.05) were 6284 and 3118, respectively. After FDR correction, it became 967 versus 209, respectively. The sliding-window cutoff method could catch more potential candidates which have *p* values far less than 0.001 for subsequent Cox’s modeling. That is because of the properly selected cutoff improving the statistical significance. A smaller *p* value might predict large effect size (HR) associated biomarkers. Then, these preliminary candidates will have an opportunity to be carefully selected by using FDR, stringent Bonferroni correction, their effect sizes (Cox’s HR), and the use of another independent cohort (GSE65858) to prevent false discovery.

We can explain the aforementioned situation by examples. When reviewing the special cases of genes such as NDFIP1, DKC1, PNMA5, and NPB, we noticed that NDFIP1, with a *p* value of 0.05 at the 50% quantile (median) cutoff, could achieve a *p* value of 2.62 × 10^−6^ at a 70% quantile cutoff. NDFIP1 has a FDR-adjusted *p* value of 1.07 × 10^−4^ (Appendix A, a “S” or “W”-shaped portion of the *p* value plot). However, it was excluded as a candidate by a small effect size (HR = 1.33 in GSE65858) of less than 1.8.

The other example is IL19. It has a *p* value plot with acute S-curve bending at the median zone, which lets the FDR-adjusted *p* value have a large difference between the 50% quantile cutoff (FDR-KM *p* = 0.115) and an optimal 48% quantile cutoff (FDR-KM *p* = 6.54 × 10^−6^). This optimal cutoff method could boost its statistic significance to pass correction by both FDR and Bonferroni methods. Even IL19 became a candidate through its effect size (HR = 0.472 in TCGA cohort), but failed in the validation with the GSE65858 cohort (small effect size as HR = 0.630, and FDR-KM *p* = 0.031).

### 3.4. Technical Considerations

There are two essential points of biomarker discovery from survival analysis of the TCGA HNSCC dataset.

First, since TCGA genomic data were harmonized, the pre-processing of TCGA RNA-Seq in our workflow was done as follows:HNSCC samples without complete clinical information were removed;Null expressed genes in more than 30% of the HNSCC samples were excluded;The updated number of protein-coding genes in the TCGA HNSCC was 20,500.


After investigation of the mRNA expression dataset obtained through NCI’s Firehose API, we found that the expression values of two genes (gene ID: 9906 and gene ID: 728661) were saved together under the entity of gene symbol “SLC35E2.” The expression file of SLC35E2 was almost double those of SLC35E1 and SLC35E3 in size. According to the Human Gene Database (available at https://www.genecards.org/Search/Keyword?queryString=SLC35E2, accessed on 10 March 2021), SLC35E2A (Gene ID: 9906) and SLC35E2B (Gene ID: 728661) should be the correct entities for the TCGA HNSCC dataset. SLC35E2 is the previous symbol of SLC35E2A (reference at https://www.genenames.org/data/gene-symbol-report/#!/hgnc_id/HGNC:20863, accessed on 10 March 2021). Thus, we reassigned the expression values of SLC35E2A and SLC35E2B and updated the number of protein-coding genes in this TCGA HNSCC dataset from 20,499 to 20,500.

Second, we analyzed the error log during the cutoff finding and Cox modeling. The result shows that program could be halted under several technical situations. These included:If 32.2% of events had a “one group” issue in the confusion matrix of Chi-square test in Cox regression (coxph), due to a zero in the M (distant metastasis) patient subgroup;If 21.05% of errors occurred via “one group” issues in log-rank test (survdiff or survdiff.fit function in R package “survival”) in the Kaplan–Meier estimate;If 0.78% had unknown reasons (so those 159 genes were excluded in our workflow).

These technical problems could not be detected prior to program running. They might have been due to skewed distribution of the expression value or even random error derived during the RNA sequencing procedure.

### 3.5. Limitations of the Study

The validation rate in the current study was 3 out of 20 (15%). The TCGA HNSCC and GSE65858 cohorts have similar demographic features. However, the percentages of females were 26.9% and 17.0%. Moreover, the percentages of smokers and tobacco they had were 76.9% and 82.2%, and 45.8 and 28.3 pack-years, respectively. In total, 60.3% of females were smokers in the TCGA HNSCC; 67.3% of females were smokers in the GSE65858. The results of the current study also show that higher tobacco exposure is an independent risk factor of HNSCC patient survival (HR 1.364 (95% CI: 1.008–1.844, *p* = 0.044)). This implies that even though gender is not a prognostic factor, the heavy female smoker might contribute some genetic alterations to HNSCC.

American white people were 85.6% of the TCGA HNSCC cohort, and German people were 90% of the GSE65858 cohort. Grunwald and his colleagues [78] found that oropharyngeal cancer is more common in North America (51.2%) than Northern Europe (32.4% Germany). However, we found only nine patients with cancer of the oropharynx in the TCGA HNSCC. Of oropharyngeal SCC patients (102 samples, 37.8%), 52.2% were positive for HPV in GSE65858. Thus, the oropharyngeal SCC was far less prevalent (1.7%) in the TCGA cohort than in the GSE65858 cohort (37.8%). The oropharyngeal SCC should have a different genetic background than non-oropharyngeal HNSCC [79]. This could be another reason for the poor consensus across different HNSCC datasets. Dataset selection and proper demographic matching are important in biomarker discovery.

Our approach was to build a regression model to predict a patient’s survival by the TCGA’s gene expression. The model was validated by GSE65858 cohort. Thus, we performed made a head-to-head comparison of Cox’s hazard ratios (5404 genes) from TCGA HNSCC and GSE65858 datasets (Figure 6a). It showed a poor [80] Pearson’s correlation coefficient [81] (r = 0.27). Figure 6b shows that 20 genes, including three biomarker candidates—CAMK2N1, CALML5, and FCGBP—have a moderate [80] effect size (HR) correlation between the two cohorts (Pearson’s r = 0.68). Overfitting of Cox’s regression model could achieve only a moderate correlation. In such a case, an overfitted model will only perfectly match every single gene in the TCGA, and has higher variability when predicting what it never saw—the GSE65858 dataset. Those, as mentioned earlier, might be the reasons why the validation rate was so low. Overfitting will limit the usefulness of the model in its generalization. It might be controlled well by using cross-validation.

Moreover, the head-to-head comparison of Kaplan–Meier *p* values (5404 and 20 genes) from those two cohorts is also shown in Appendix A. It reveals poor Pearson’s correlations (r = 0.01 and r = 0.19). Thus, the significance of CAMK2N1, CALML5, and FCGBP was still sufficient compared to that of the other 17 genes in the GSE65858 study. There were 270 participants whose data were collected in GSE65858 for our validation study. Again, the effect size (i.e., hazard ratio) can be large but fail to gain statistical significance if the sample size is not large enough. A high-quality HNSCC dataset (with protein-coding gene expression and survival data, *n* > 500) is not easy to find in the GEO database. The three biomarker candidates were discovered by TCGA HNSCC and then validated by the GSE65858 dataset. More data are required for further confirmation.

### 3.6. Future Directions in Translational Medicine

#### 3.6.1. Proteomics Validation

Although we combined the power of genome-wide scanning and an optimal cutoff finder for survival analysis, the study has some limitations. We are aware of the importance of direct assessment of protein products comprising the basic functional units in cancer cells’ biological processes. The announcement of the Cancer Proteome Atlas (TCPA, http://tcpaportal.org, accessed on 10 March 2021) excites the cancer research community [82,83]. Through the utility of the reverse-phase protein arrays (RPPAs) or reverse-phase protein lysate microarray (RPMA), microarray “Western blots” in the TCPA could help to test our hypotheses from RNA-Seq studies. However, in the TCPA database (v3.0 [84]), there are only 237 antibodies available, not covering our candidates so far.

#### 3.6.2. Laboratory Validation

We encourage multidisciplinary studies that use complementary computational and experimental approaches to address challenging cancer research. Such in vitro and in vivo validation experiments will be undertaken in our laboratory. We plan to analyze the mRNA (e.g., qRT-PCR) and protein (e.g., Western blot) of HNSCC cell lysate to confirm the candidate genes’ expression. The effects of overexpression and knockdown of the genes by lentiviral clones should be observed in cell function assays (e.g., proliferation, migration, and invasion) and mouse xenograft models (e.g., tumor growth).

Moreover, this bioinformatics paper provides targets and supports the community’s rationale for looking into these HNSCC candidates via in vitro and in vivo validation. We aim to promote a reproducible bioinformatics [85,86] method allowing successful repetition and extension of analyses based on the TCGA or other in-house HNSCC datasets. Good research reproducibility practice is necessary to allow the reuse of code and results for new projects. It may turn out to be a time-saver in the longer run. When multiple scientists can reproduce a result, it will also validate our initial results and readiness to progress to the next research phase. Once our laboratory or the community confirms those candidates as targets, compound screening [87,88,89] could facilitate more personalized therapy for HNSCC patients.

#### 3.6.3. Cancer Type-Agnostic Study

Our strategy still has the strength to explore more possible biomarkers from RNA-Seq datasets in cancer research. In our previous work, altered glucose metabolism—the Warburg effect [90]—promoted the progression of HNSCC, which is partially attributed to the solute carrier family 2 member A4 (SLC2A4, or glucose transporter 4, GLUT4) and tripartite motif-containing 24 (TRIM24) pathway [91,92]. Lactic acidosis-induced GLUT4 overexpression was also found in lung cancer cells [93]. Currently, pembrolizumab and nivolumab’s success has been based on a common biomarker (e.g., PD-1) in several types of cancer. It is a model of tumor type-agnostic therapy [31]. There are several common biomarkers of immune-checkpoint inhibitor (ICI) under evaluation, including tumor-infiltrating lymphocytes (TIL), interferon gamma (IFN-γ), and tumor mutational burden (TMB) [23]. The other ICI, anti-LAG-3 (pelatlimab), is currently being evaluated in phase I/IIA [50] (ClinicalTrials.gov Identifier: NCT01968109) and II-IVA [94] (NCT04080804) studies.

In line with tumor-agnostic research, we plan to explore common biomarkers crossing TCGA diseases. However, the GDC provided standardized data frames that could not directly fit our workflow’s scope. Before the global gene scanning process, it is necessary to re-format, transpose, and merge the 528 patients’ clinical datasets and correlate 20,500 expressions of bio-specimens. This process should be carefully curated to confirm the data integrity within the correct definition [95]. We also plan to upgrade our R script for the cutoff engine to C++ and source it in the Rstudio server. The high performance of C++ could speed up the critical steps in this workflow involving heavy computation of matrix data [96]. Moreover, it will be possible to introduce the Rstudio Shiny app (https://shiny.rstudio.com, accessed on 10 March 2021) as a web-based tool (named “pvalueTex”) packaged with our workflow in the future.

#### 3.6.4. Holistic Cancer Care

There are 81 physical, pathological, and social conditions derived from participants available for survival modeling in the TCGA, such as age, gender, residual tumor, vital status, days-to-last-followup, cancer stage, smoking duration, exposure to alcohol, asbestos, and radioactive radon. However, the TCGA did not collect other features related to holistic care. When going for holistic cancer care [97,98], spiritual and emotional conditions are equally essential, alongside physical and social status (Figure 7). Psychosocial stress is associated with cancer incidence [98,99,100], metastasis [99,101,102,103], and poor survival [104]. These impacts might be mediated through the hypothalamic–pituitary–adrenal (HPA) axis [105]. Holistic healthcare providers engage patients with eye contact for mind-to-mind connection. Their empathy, sympathy, and compassion are induced by the suffering of patients from those diseases. They try to treat patients by prescribing medicine (and “themselves”) or performing surgery. Thus, the healing resilience of patients should be induced by unconditional positive regard. The patients trust those who take care of them and have the confidence to increase the capacity to recover from diseases through a mind–brain–body connection manner (Figure 7).

## 4. Materials and Methods

### 4.1. Patient Cohort

A large-scale cancer database, aggregating many independent features, is necessary to facilitate the biomarker discovery. The Cancer Genome Atlas (TCGA) project [106] has been developed since 2005 and supervised by the National Cancer Institute’s (NCI) Center for Cancer Genomics and the National Human Genome Research Institute (NHGRI), funded by the US government. TCGA represents comprehensive genomics and clinic data from 84,392 patients among 33 major cancer types (data release 27.0—29 October 2020, available at https://www.cancer.gov/about-nci/organization/ccg/research/structural-genomics/tcga/studied-cancers, accessed on 10 March 2021). TCGA and the genome data analysis center (GDAC) generated and analyzed DNA (mutations, copy number variations, methylation sites, simple nucleotide polymorphisms), RNA (microarray, RNA-Seq, microRNA), and protein (reverse protein phased array) data derived from biospecimens. Sample types available at TCGA are primary solid tumors, recurrent solid tumors, blood-derived normal and tumor, metastatic, and solid normal tissue.

The NCI’s Genomic Data Commons (GDC, available at https://portal.gdc.cancer.gov, accessed on 10 March 2021) receives, processes, and distributes genomic, clinical, and biospecimen data from the TCGA database and other cancer research programs. The clinical features have been defined by TCGA GDC data dictionary: Common Data Element (CDE) [107]. The RNA-Seq expression data have been harmonized and re-aligned against an official reference genome build (Genome Reference Consortium Homo sapiens genome assembly 38, GRCh38). TCGA, GDC, and some research communities offer several computational tools to the public for facilitating cancer research. GDC Data Portal has the official web-based TCGA data analysis tools. Other available web-based tools have been reviewed by Zhang et al. [108] and Matthieu Foll (availalbe at https://github.com/IARCbioinfo/awesome-TCGA, accessed on 10 March 2021). One of the GDACs, the Broad TCGA Data and Analyses (Broad GDAC), provides Firehose, a repository of the TCGA public-accessible Level 3 data and Level 4 analyses. Broad GDAC Firehose is an analytical infrastructure that analyses algorithms not performed by the GDC (e.g., GISTIC, MutSig2CV, correlation with clinical variables, mRNA clustering). A web-based version of Broad GDAC Firehose is Firebrowse (available at firebrowse.org, Version: 1.1.40, 13 October 2019). Broad GDAC Firebrowse provides graphical tools such as viewGene to explore expression levels and iCoMut to explore a mutation analysis of each TCGA disease.

GDC’s application programmable interface (API) uses the Representational State Transfer (REST) architecture and provides accessibility to external users for programmatic access to the same functionality found through GDC Portals. Those functions include searching, viewing, submitting, and downloading subsets of data files, metadata, and annotations based on specific parameters. Moreover, if restricted data are requested, the user must provide a token along with the API call. This token can be downloaded directly from the GDC Portals. Broad GDAC Firebrowse RESTful API can be accessed using an R package, FirebrowseR (available at https://github.com/mariodeng/FirebrowseR, accessed on 10 March 2021) [109].

GDC is available at https://portal.gdc.cancer.gov/projects/TCGA-HNSC, accessed on 10 March 2021. TCGA offers several computational tools to the public that facilitat cancer research. Broad genome data analysis center (GDAC) Firebrowse (firebrowse.org, version 1.1.35, 27 September 2016) is one of those tools to provide data access for each TCGA disease through a Representational State Transfer (REST) application programmable interface (API). The 528 TCGA HNSCC patients’ clinical information and RNA-Seq data were obtained from the Firebrowse RESTful API with an R package, FirebrowseR (available at https://github.com/mariodeng/FirebrowseR, accessed on 10 March 2021) [109]. We utilized FirebrowseR with our analysis workflow (Figure 1, green square) to receive standardized data frames while avoiding data re-formatting, often causing some errors. GSE65858 is a dataset we used for candidate selection in our workflow. After removing missing data, there were 270 participants whose data were collected for our validation study. Initially, there were 288 HNSCC participants involved in their prospective study [51]. At the University Hospital Leipzig, Germany, these patients were diagnosed as having oral, oropharyngeal, hypopharyngeal, or laryngeal squamous cell carcinomas (SCCs). Patients were excluded if they had a prior history of cancer other than HNSCC within the last five years. The 49 (17.0%) females and 239 (83.0%) males had a median age of 58 years old. A total of 82.2% were smokers who consumed 28.3 pack-years of cigarettes. In total, 88.5% of participants used alcoholic beverages; 84.9% with oral SCC were HPV-negative; 52.2% with oropharyngeal SCC were positive for HPV. The cancer stage distribution among this cohort was 19.0% early stages (I/II) and 81.3% late stages (III/IV).

Regarding the TCGA database, 528 HNSCC participants from several centers were used in the prospective studies [110]. The 142 (26.9%) females and 386 (73.1%) males had a median age of 61 years old. A total of 97.5% were smokers who consumed 45.8 pack-years of cigarettes. In total, 67.6% of participants used alcoholic beverages; 82.1% of participants with oral SCC were HPV-negative. The cancer stage distribution among this cohort was 104 (20.7%) early stages (I/II) and 424 (79.3%) late stages (III/IV).

#### 4.1.1. RNA Sequencing Data

The number of protein-coding genes was suggested to be 20,500 [111]. The GDC Data Portal-provided TCGA data were harmonized with re-aligned RNA sequencing data against an official reference genome build (Genome Reference Consortium Homo sapiens genome assembly 38, GRCh38). RNA-Seq expression level read counts produced by Illumina HiSeq were normalized using the Fragments per kilobase per million reads mapped (FPKM) method, as described in [112]. The RNA-Seq preprocessor of Broad GDAC picked the RNA-Seq by Expectation-Maximization (RSEM) value from Illumina HiSeq/GA2 messenger RNA-Seq level_3 (v2) dataset of NCI GDC. It made the messenger RNA-Seq matrix with log2 transformed for the downstream analysis, as described in their paper [113]. We utilized FirebrowseR’s function call, Samples.mRNASeq(cohort = “HNSC,” gene = GeneName, format = “csv”), to download the RNA-Seq dataset of every HNSCC patient and to save 20,499 data frame files, named “HNSCC.mRNA.Exp.[GeneName].Fire.Rda.” After careful investigation of the genomics dataset, the RNA-Seq values of “solute carrier family 35 member E2A (SLC35E2A)” and “solute carrier family 35 member E2B (SLC35E2B)” were considered two distinct expression entities. We concluded that the number of protein-coding genes in the TCGA dataset is 20,500. We removed null expressed genes, over 30% of the cohort, to avoid useless results.

#### 4.1.2. Clinical Data

We utilized FirebrowseR’s function call, Samples.Clinical(cohort = “HNSC,” format = “csv”), to get all 81 clinical features (including pathological data, defined by TCGA GDC data dictionary: Common Data Element (CDE) [107]) of all 528 HNSCC patients, which were saved as one data frame file: “HNSCC.clinical.Fire.Rda” (accessed November 2019).

“HNSCC.clinical.Fire.Rda” tables each have 20,500 “HNSCC.mRNA.Exp.[GeneName]. Fire.Rda” tables were transposed and merged by their _participant_barcode (unique patient identification, ID) to yield a data frame with 528 rows (participants) against 20,581 columns (81 clinical features and 20,500 protein-coding RNA-Seq of cancer specimens). The clinicopathological features selected for our workflow included gender, age, clinical tumor size, clinical cervical lymph node metastases, clinical distant metastasis assessment, pathological surgical margin, and tobacco exposure with their corresponding survival data. The tumor size (T), cervical lymph node metastases (N), and distal metastasis status (M) were classified according to the American Joint Committee on Cancer (AJCC) [62] along with he Union for International Cancer Control (UICC) [114] TNM system for clinical staging of HNSCC. We made data clean by removing duplicated rows and columns.

### 4.2. Cutoff Finder Core Engine

To evaluate the effect of gene expression on patient survival, we used sliding-window cutoff selection by stratifying patients with Kaplan–Meier survival analysis according to each gene’s low/high expression. Our cutofFinder_func subroutine employs the minimum *p* value approach to recognizing cutoff points in continuous gene expression measurement for patient sub-populations. First, patients were ordered by RNA-Seq values (RSEM) of a given gene. Next, patients were stratified at a serial cut (counted people ranked between the 30th and 70th percentiles of the cohort; Figure 1 cutoff engine). The survival risk differences of the two groups were estimated by log-rank test to yield around 165 Kaplan–Meier *p* values for each gene. Then, the optimal cutoff of RNA-Seq giving the minimum *p* value was selected by the cutofFinder_func subroutine. This iteration method could calculate all possible cutoffs of each gene’s expression in this cohort. After each run of the cutofFinder_func function call for an individual gene, it returned an optimal cutoff for specific patient groups (e.g., low expression in 262 persons versus high expression in 152 persons with calcium/calmodulin dependent protein kinase II inhibitor 1; Figure 5). The cutoff would be returned to the main program to allow downstream Cox survival analysis. The percentile range we applied, 30% to 70%, was used to avoid a small grouping effect [47,115]. In case there was no significant *p* value, a median expression of this gene was set as its cutpoint as usual. The false discovery rate (FDR) (<0.05) correction [116] shows which genes should be retained for subsequent univariate and multivariate analysis. It ensures the control of type I error of multiple tested *p* values during our cutoff finding procedure. Then Bonferroni adjustment of that *p* values was used for candidate selection.

### 4.3. Statistical Consideration for Survival Analysis

Our workflow has loops to call the function survival_marginSFP(GeneName) with the given GeneName to process the survival analysis gene by gene. We dichotomized the clinicopathological features, which included gender (male/ female), age at diagnosis (below/beyond 65 years-old), clinical tumor size (T1-2/T3-4), clinical nodal status (negative/positive), clinical distant metastasis (negative/positive), TNM staging (early/late), surgical margin status (negative/positive), and tobacco exposure (low/high). The patients were grouped by an RNA-Seq value of each gene—low or high-expression according an optimal *p* value obtained from the cutofFinder_func subroutine (see the section of “Cutoff Finder Core Engine”). Pearson’s chi-square test was used for these binary variables. Kaplan–Meier survival was analyzed using the log-rank test for two groups OS comparison.

The Cox proportional-hazards regression model [117,118] is commonly used for modeling survival data. It allows analyzing survival for one or more variables and provides the effect sizes (coefficients, i.e., hazard ratios) for them [119]. The Cox model also accounts for confounding factors [120]. It is expressed by the hazard function denoted by *h*(*t*). The hazard function represents the risk of a specific event (e.g., death) at time *t*. It can be estimated as follows:$$h\left(t\right)=h_{0}\left(t\right)\times exp\left(\beta_{1}X_{1}+\beta_{2}X_{2}+\beta_{3}X_{3}+...+\beta_{n}X_{n}\right)$$
where*t* represents the survival time;h(t) is the hazard function determined by a set of *n* covariates ($X_{1}...X_{n}$)—e.g., clinicopathological features, including age, gender, gene expression, cancer stage (tumor size, nodal metastases, distant metastases), surgical margin, smoking, and alcohol; unfortunately, spiritual, emotional, and social status are not available in TCGA database;The coefficients ($\beta_{1}...\beta_{n}$) measure the impacts—the effect sizes of covariates;The term $h_{0}$ is called the baseline hazard. It corresponds to the hazard value if all the $X_{i}$ are equal to zero. The “*t*” in h(t) indicates the hazard may vary over time.


Thus, the biomarker discovery strategy is survival modeling through a collection of $X_{1}...X_{n}$ features from cancer datasets.

A univariate Cox proportional regression model, using the “coxph” function in R package “survival,” has been applied to calculate hazard ratios, 95% confidence interval (95% CI), and significance, and to estimate the independent contribution of each clinicopathological feature and gene expression level to the overall survival.

In a multivariate test, those covariates used include the clinicopathological features (gender, age at diagnosis separated by being 65 years old or not, clinical tumor size (T1 or T2/T3 or T4), clinical nodal status (N0/N+), clinical distant metastasis (M0/M1), TNM staging (stage 1 or 2/stage 3 or 4), surgical margin status (negative/positive), and tobacco exposure (low/high)); and gene expression levels (low/high) defined by cutoffs. All covariates were pooled in the hazard function *h*(*t*) to estimate their impact on the overall survival.

Results were considered statistically significant when a two-sided *p* value was less than 0.05, or a lower threshold if indicated. The false discovery rate (FDR) (<0.05) could be used to pick up the optimal *p* value to ensure the control for type I error of the Kaplan–Meier survival test during the cutoff finding procedure. There were also multiple correlated tests of null hypotheses during our global scanning of 20,500 protein-coding genes. The stringent Bonferroni correction could result in an adjusted *p* value to ensure the control for type I error.

The resulting data, including Kaplan–Meier curves, cumulative *p* value plots, and Cox regression tables, were exported to “.xlsx” and “.Rda" files (by R package “r2excel”) for subsequent biomarker selection.

### 4.4. Biomarker Selection and Validation

Those genes with prognostic impacts, whose hazard ratios were >=1.8 or <=0.6 in both Cox models (univariate and multivariate), were assigned as provisional candidates. Bonferroni-adjusted (Kaplan–Meier) *p* values were used to rank candidates for the decision (Figure 1, candidate selection).

GSE65858 [51] is a dataset we used for helping with candidate selection in our workflow. The Gene Expression Omnibus (GEO) database [121], founded by National Center for Biotechnology Information (NCBI), is a public repository supporting MIAME-compliant data, including microarray and sequence-based experiments. The GEOquery R package [122] was used to download the RNA-Seq dataset (in SOFT or MINiML format) of a HNSCC cohort, GSE65858, from the GEO database (available at https://www.ncbi.nlm.nih.gov/geo/geo2r/?acc=GSE65858, accessed on 10 March 2021). GSE65858 has OS, RFS, and survival time. There were 270 HNSCC participants involved in this cohort. The expression data were generated using the platform GPL10558 (Illumina HumanHT-12 v4.0 Expression BeadChip), which targets more than 30,330 annotated genes (47,000 probes, derived from the NCBI Reference Sequence, release 38 on 7 November 2009). We have performed Kaplan–Meier (with FDR-correction of *p* value) and Cox survival analyses with gene expression cutoffs at their median values. The biomarker candidates were a consensus result of TCGA and GSE65858 analyses.

## 5. Conclusions

Our findings suggested three biomarker candidates—CAMK2N1, CALML5, and FCGBP—which are all heavily associated with the prognosis of OS under an optimal cutoff with stringent Bonferroni *p* values and proper effect size (HR).

The microenvironment of HNSCC, influenced by the mind–brain–body axis, requires further exploration and understanding using holistic multi-parametric approaches. Since mindfulness meditation will be helpful in cancer healthcare, we continually educate our cancer patients that they should confess for not taking care of their bodies and spirits in the past, and give sincere thanks for their physical body’s hard work.

## Figures and Tables

**Figure 1 jpm-11-00782-f001:**
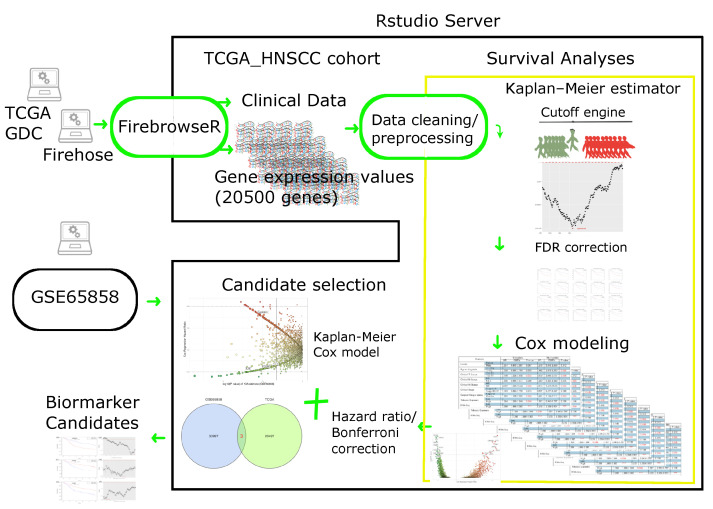
A workflow of HNSCC biomarker discovery. The workflow includes data retrieval from the TCGA GDC data portal, data processing with merging and cleaning, and then performing the survival analyses (within yellow square). The Cutoff engine (in R script: cutofFinder_func.HNSCC.R, a serial cutoff for grouping patients with low or high expression of a specific gene, to yield a collection of *p* values; please see Materials and Methods section for details) might calculate all possible Kaplan–Meier *p* values (corrected by false discovery rate (FDR) method) to find the optimal cutoff value of gene expression for subsequent Cox modeling. The candidate selection performs (1) dissection and selection of candidate genes with further Bonferroni-adjusted *p* values and the hazard ratios of a Cox model, based on the results from the survival analyses; (2) survival analyses of the other HNSCC dataset (GSE65858) using Kaplan–Meier estimates (with FDR corrections) and Cox modeling. The biomarker candidates were consensus results of TCGA and GSE65858. (HNSCC: head and neck squamous cell carcinoma; TCGA: the Cancer Genome Atlas; RNA-Seq: RNA sequencing; GDC: Genomic Data Commons).

**Figure 2 jpm-11-00782-f002:**
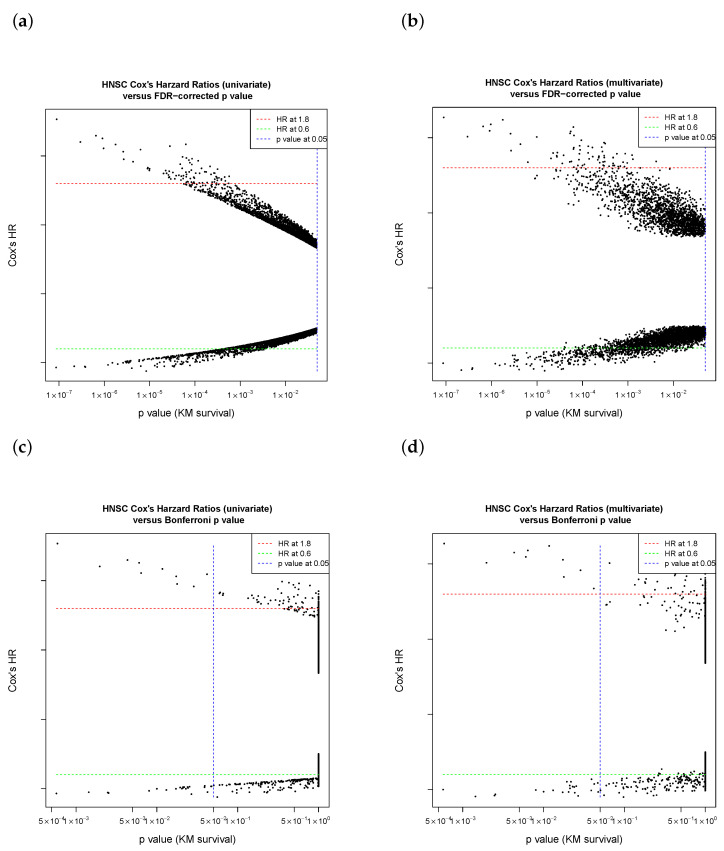
The initial progress of candidate selection from the TCGA HNSCC cohort. The *p* value of Kaplan–Meier survival was one of the selection criteria. The effect size was estimated by Cox’s hazard ratio. Initial trial step: (**a**) univariate HR versus FDR-adjusted *p* value; (**b**) multivariate HR versus FDR-adjusted *p* value. After stringent restriction by Bonferroni-adjusted *p* values and Cox’s HR, a few top-ranked genes were acquired by (**c**) univariate HR versus Bonferroni-adjusted *p* value; (**d**) multivariate HR versus Bonferroni-adjusted *p* value. (TCGA: the Cancer Genome Atlas; HR: hazard ratio; FDR: false discovery rate).

**Figure 3 jpm-11-00782-f003:**
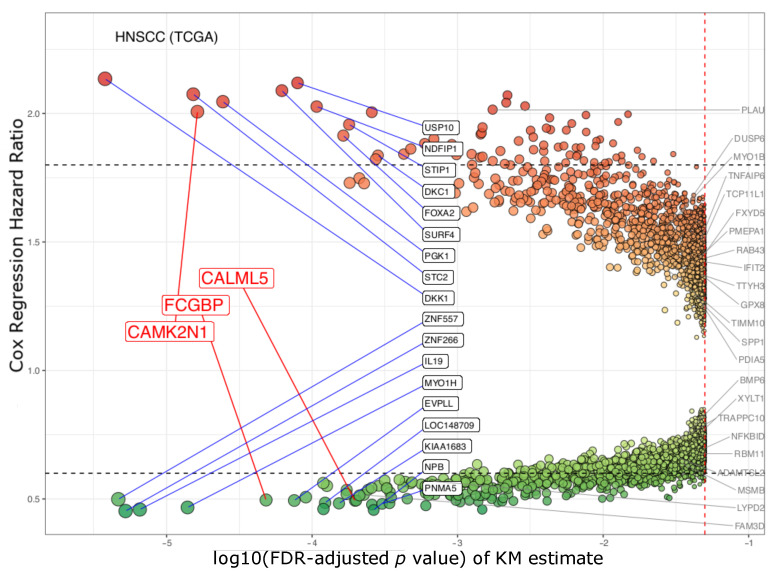
A volcano plot of genes in survival analyses of TCGA HNSCC. This cohort was applied for exploration of the candidate biomarkers. A total of 9416 genes had unadjusted *p* values of less than 0.05. CAMK2N1, CALML5, FCGBP, and 17 other genes (marked in black square) had hazard ratios (HRs) >1.8 or <0.6. The 22 genes, listed on the side, had hazard ratios between 0.6 and 1.5. Red spots: HR > 1.0. Green spots: HR < 1.0. (X-axis: Kaplan–Meier survival estimates, with FDR-adjusted *p* values (log10 transformed); y-axis: HR of Cox proportional hazard regression model.)

**Figure 4 jpm-11-00782-f004:**
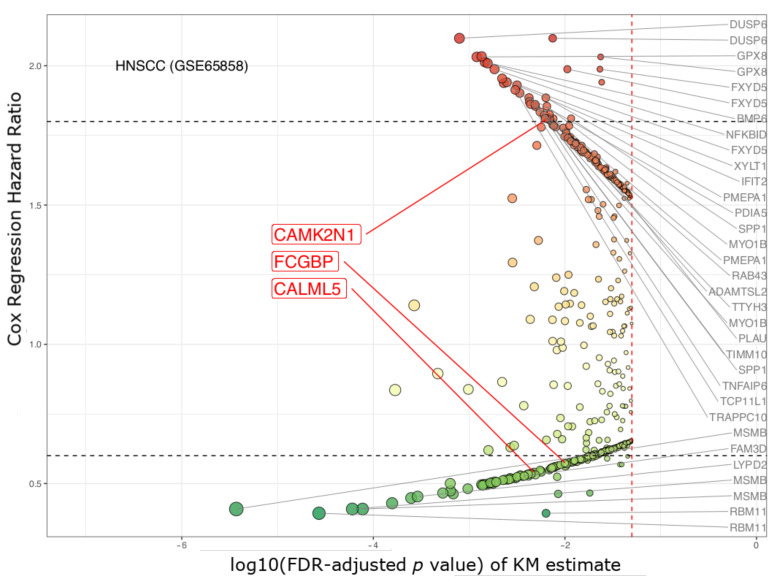
Volcano plot of genes in survival analyses of GSE65858 cohort. This HNSCC cohort was used for filtering of our candidate genes: CAMK2N1, CALML5, and FCGBP. In total, 534 genes had FDR-adjusted *p* values less than 0.05 Red spots: hazard ratios are greater than 1.0; Green spots: hazard ratios are under 1.0. The 22 genes, listed on the side, had hazard ratios >1.8 or <0.6. (X-axis: Kaplan–Meier survival estimates, with FDR-adjusted *p* values, log10 transformed; y-axis: the hazard ratio (HR) under the Cox proportional hazard regression model).

**Figure 5 jpm-11-00782-f005:**
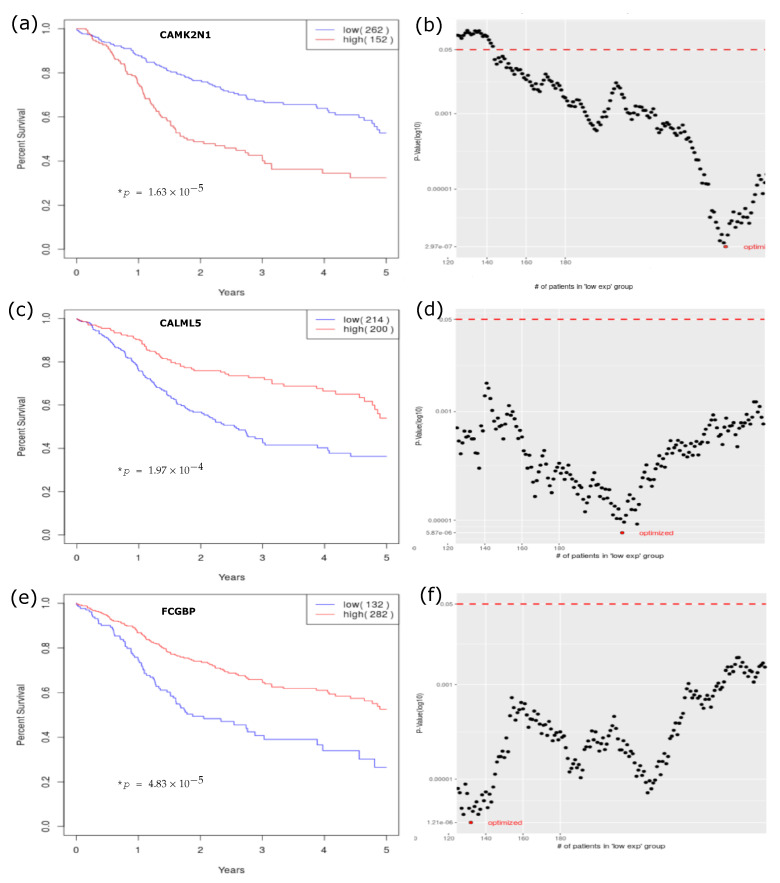
Kaplan–Meier survival analyses, by cutoff finding. The Kaplan–Meier curves of (**a**) CAMK2N1, (**c**) CALML5, and (**e**) FCGBP with optimal *p* values. The cutoffs in the cumulative *p* value plots of (**b**) CAMK2N1, (**d**) CALML5, and (**f**) FCGBP, show that over 50% of those unadjusted *p* values derived by the sliding-window cutoff-finding procedure are below 0.001. (* *p*: *p* value adjusted by false discovery rate, FDR).

**Figure 6 jpm-11-00782-f006:**
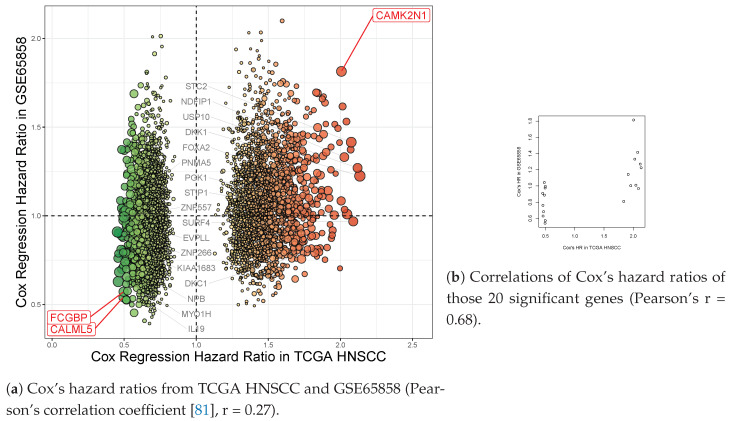
A head-to-head comparison of Cox’s hazard ratios from TCGA HNSCC and GSE65858 datasets. TCGA HNSCC and GSE65858 cohorts were applied for identification and validation of the candidate biomarkers in HNSCC. (**a**) A total of 5404 genes had Cox’s hazard ratios from TCGA HNSCC and GSE65858 (Pearson’s correlation coefficient [81], r = 0.27). CAMK2N1, CALML5, FCGBP, and 17 other genes (marked in black) had hazard ratios (HRs) >1.8 or <0.6. Red spots: *HR*s > 1.0 in TCGA HNSCC. Green spots: *HR*s > 1.0 in TCGA HNSCC. Sizes of spots: bigger for Kaplan–Meier p values in TCGA HNSCC. (**b**) The 20 genes were extracted and shown. The hazard ratios of those genes have a moderate correlation between the two cohorts (Pearson’s r = 0.68). (X-axis: Hazard ratios of Cox proportional hazard regression model from TCGA HNSCC; y-axis: Those values from GSE65858; TCGA: the Cancer Genome Atlas; HNSCC: head and neck squamous cell carcinoma).

**Figure 7 jpm-11-00782-f007:**
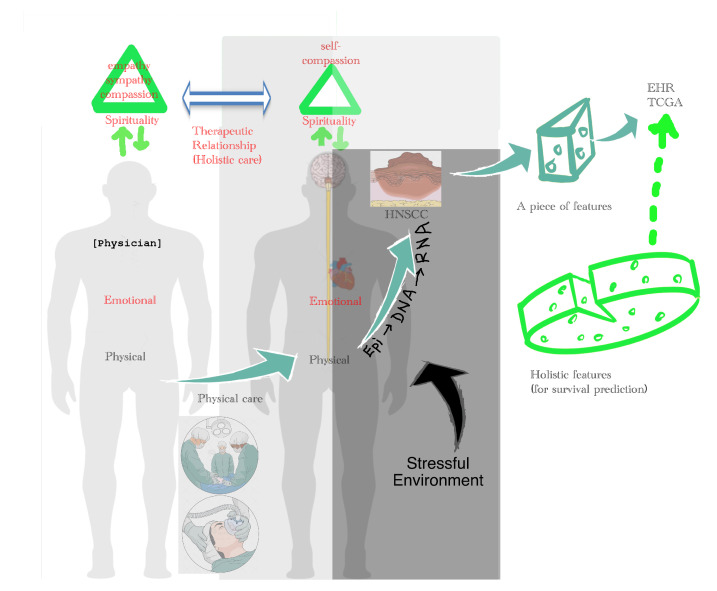
The concept of holistic care for HNSCC patients. Beyond carcinogenesis: In the mind–brain–body axis, a stressful environment (giant black arrow) will trigger an emotional response. The subconscious mind (brain) releases stress hormones and inflammation signals in response to negative emotions. The body’s internal environment (cells) alters epigenetic control in gene regulation and mRNA expression. Over a long time, the tissue/cells will be transformed into dysplasia and then malignancy (e.g., HNSCC) with help from known carcinogens. Cancer care: Holistic care should take care of cancer patients’ spiritual, emotional, physical, and socioeconomic needs. Physical care will be carried out by medication therapy or surgery. After establishing a therapeutic relationship (TR), the physicians’ spiritual properties (empathy, sympathy, and compassion) will engage cancer patients and recover their self-compassion to gain resilience against the disease through their mind–brain–body axis. Thus, we suggest that electric healthcare records (EHR) should include physical, pathological, and psychological data, and even more spiritual information. The TCGA might collect those “holistic features” (green dashed line) for further study of personalized medicine.

**Table 1 jpm-11-00782-t001:** Univariate/multivariate Cox proportional hazard regression analyses on OS time of CAMK2N1 gene expression in HNSCC.

Features	Univariate			Multivariate			
		HR	CI95%	*p* Value	HR	CI95%	*p* Value
Gender	Female	1			1		
	Male	1.157	0.843–1.587	0.367	1.076	0.767–1.510	0.671
Age at diagnosis	<=65y	1			1		
	>65y	1.329	0.990–1.784	0.058	1.391	1.025–1.888	0.034
Clinical T Status	T1 + T2	1			1		
	T3 + T4	1.409	1.028–1.931	0.033	1.982	1.048–3.745	0.035
Clinical N Status	N0	1			1		
	N1-3	1.185	0.890–1.577	0.246	1.145	0.801–1.636	0.457
Clinical M Status	M0	1			1		
	M1	4.097	1.009–16.644	0.049	7.314	1.590–33.631	0.011
Clinical Stage	Stage I + II	1			1		
	Stage III + IV	1.245	0.882–1.759	0.213	0.621	0.287–1.343	0.226
Surgical Margin status	Negative	1			1		
	Positive	1.591	1.155–2.191	0.004	1.631	1.182–2.250	0.003
Tobacco Exposure	Low	1			1		
	High	1.364	1.008–1.844	0.044	1.363	0.990–1.875	0.058
Gene Expression	Low	1			1		
	High	2.101	1.572–2.809	< 0.001	2.007	1.490–2.704	<0.001

(OS: overall survival; HR: hazard ratio; CI95%: 95% confidence interval; *p* value significant code is denoted: red < 0.05).

**Table 2 jpm-11-00782-t002:** The top 3 genes with prognostic impacts on HNSCC.

Gene ID	Gene Description	Kaplan–Meier Survival		Cox Univariate		Cox Multivariate	
		FDR *p* Value	Bonferroni *p* Value	HR *	CI95%	HR *	CI95%
CAMK2N1	calcium/calmodulin- dependent protein kinase II inhibitor 1	1.63 × 10^−5^	0.002	2.101	1.572–2.809	2.007	1.490–2.704
CALML5	calmodulin like 5	1.97 × 10^−4^	0.039	0.51	0.379–0.686	0.493	0.364–0.667
FCGBP	Fc fragment of IgG binding protein	4.83× 10^−5^	0.008	0.484	0.359–0.653	0.496	0.366–0.674

Selection criteria (fit all): (1) Kaplan–Meier Bonferroni-adjusted *p* < 0.05; (2) Cox’s univariate and multivariate HR >= 1.8 or <= 0.6 in TCGA cohort; (3) Cox’s univariate and multivariate HR >= 1.8 or <= 0.6 in GSE65858 cohort. * Cox’s model: *p* <0.001 (HR: hazard ratio; CI95%: 95% confidence interval; FDR: false discovery rate).

## Data Availability

All data processing and analyses were performed with R programming language (https://www.r-project.org/, version 4.0.2, 22 June 2020) and R packages “firebrowseR,” “survival,” “reshape,” “data.table,” “ggplot2,” “R.utils,” “xlsx,” “r2excel,” “rJava,” and “rms” in an Rstudio server (version 1.2.5001) based on Google cloud platform in Linux (Ubuntu LTS, release v18.04.3). The R script code and datasets generated during the current study are available at the GitHub repository, https://github.com/texchi2/pvalueTex, accessed on 10 March 2021.

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
