# Peer review of "A Transcriptomic Analysis of Head and Neck Squamous Cell Carcinomas for Prognostic Indications"

_jpm, 2021, doi:10.3390/jpm11080782_

Round 1

Reviewer 1 Report

This is a revised manuscript, whose original version  I have reviewed previously. I appreciate that the authors add a validation dataset. The validation rate is low (3 out of 20), and even the three positive genes just have moderate significance (~0.03). I am ok with the authors' strategy for analyzing TCGA data is reasonable. Then the result here questions the usability of identifying robust biomarkers using TCGA RNA-seq data and demonstrates the necessity of additional tests with independent datasets. This is super important and should be mentioned in the abstract. Please also review the validation landscape in other papers (identifying biomarkers using TCGA RNA-seq for other cancer types) and discuss why the validation rate is that low.

The validation analysis (GSE65858 dataset) should be improved. Please introduce data details (sample size, population background, data type, et al.) in the main text. The authors used a similar strategy (using a cutoff of median expression value instead of a sliding window) to analyze GSE65858 data again. Only three of the 20 previously identified markers are still identified, suggesting a validation rate of 3/20=0.15. The low accuracy suggests the limitation of the study and also highlights the necessity of testing markers using independent data. Even the three positive genes also just have adjusted p-values >0.03. This challenges the usability of these markers. Please discuss the limitations and possibly why their powers are different in the two datasets (e.g., differences in data generation techniques and population backgrounds). Please provide p-values/adjusted p-values of all the previously identified markers and scatter plots for HR calculated from both datasets.
Most important, the authors need to provide additional validation analysis, but not just analyze GSE65858 using a similar strategy. The authors should describe a diagnostic challenge, build single-gene prediction models based on TCGA data, apply each model to the validation data, and comprehensively evaluated the performance (sensitivity and specificity or AUC if a sliding window is used). The authors can also build multiple-gene models and evaluate them similarly.

Given the low validation rate, I suggest the authors test the markers with more independent data. If no additional data are available, please describe in the manuscript that even the three markers have been validated by GSE65858 dataset, more data are required for further validation.

It is not necessary to describe the metrics (e.g., p-values) of the three identified genes. I suggest the authors search the literature and discuss their roles in cancer.

The manuscript is hard to follow. Please improve the language.

Reviewer 2 Report

To obtain more accuracy in the survival rate analysis using TCGA data, the authors performed various bioinformatical and statistical attempts focusing on the cutoff point determination.

Above all, the strength of this study would be that they tried to confirm questions that are easily overlooked when analyzing numerous cancer-related data using various platforms. The traditional experimental process of finding results has been to go through reconfirmation experiments based on information obtained from papers and preliminary experiments. However, it would be expected that more accurate results can be derived through intensive experiments after first going through big-data analysis via various platforms.

In particular, multi-faceted confirmation through TCGA R-seq results, in-house proteome analysis, and protein atlas is also considered strengths of the results of this study. I would like to suggest the followings:

  1. In the analysis of R-seq data, the contents of miRNA analysis targeting the three genes, respectively, finally confirmed in this study, were omitted. By analyzing the expression of candidate miRNAs regulating these genes simultaneously, the expression level of these candidate genes will give more confidence.
  2. Please delete every 'please see' in (please see Figure... ).

Round 2

Reviewer 1 Report

The manuscript has been substantially improved. I am satisfied with the scientific results, but the new results are not very well integrated in the manuscript. 

Please generate two figures that directly compare HR and log p-value for the two datasets respectively. 
For each figure, X-axis is the HR/p-value from TCGA experiment and Y-axis is the value from the independent test. Plot all tested genes and highlight those 20 significant genes. Compute correlations for all tested genes and those significant ones.

Author Response

Please see the attachment. Thanks.

This manuscript is a resubmission of an earlier submission. The following is a list of the peer review reports and author responses from that submission.

Round 1

Reviewer 1 Report

This manuscript perhaps describes the biomarkers of head and neck SCC using several data-bases. They also describe the relationship between the expression level of each candidate biomarkers in their patients and other factors such as gender, age, margin, lymph-node metastasis and so on.

I myself think this manuscript is very interesting, however, I do not find the difference between the previously published manuscript from your institute.

https://www.researchsquare.com/article/rs-80673/v1

First of all, please describe the difference between the two manuscripts.

Secondly, the latter half of the discussion may not be appropriate for this manuscript. Many people aware of the relationship between prognosis and mental/financial/familial situations. Unfortunately, almost all authors do not understand why half of the discussion part was occupied with spiritual something. Isn’t it a totally different theme from biomarkers of HNSCC?

Stress may affect the expression level of these biomarkers, which may result in an unexpected prognosis, the main theme of this manuscript is different. If you want to describe spiritual something, please submit it as a brand new manuscript.

Please remember my opinion does not intend to insult your religion because I am also a Buddhist and morning meditation always allow me to start a wonderful day.

Reviewer 2 Report

The authors introduced a workflow to identify gene-expression biomarkers based on TCGA data and applied it to head and neck squamous cell carcinoma. The authors tried to highlight their workflow and present it as a new tool, however, their workflow is very common, and I cannot see its novelty. The selection of gene expression cutoff is not well handled statistically (see below my first comment). Therefore, I recommend the authors weaken their statement of the workflow but focus on the identified biomarkers in neck squamous cell carcinoma and use as much as possible evidence to support or validate them.

Exploring all possible expression cutoffs is controversial. A robust expression biomarker should be effective and detectable at most major cutoffs. Compared to a fixed quantile cutoff, a sliding window cutoff selection of cutoffs would capture more genes whose p-values just reach the significant level (p=0.05). For example, a gene with a p-value of 0.1 at 25% or 50% quantile cutoff could achieve a p-value of 0.04 at 70% quantile cutoff. But it is not expected that a gene with a p-value of 0.1 at 25% or 50% quantile cutoff could achieve a p-value of 1e-5 at 70% quantile cutoff. This strategy only identifies more weak “biomarkers,” and it also captures more false positives. Adjustment of multiple test p-values should be applied here, which is missing in the current manuscript.
Exploring all possible expression cutoffs is also usable but should only be used for established biomarkers. Once it is known that a gene is a strong biomarker (e.g., from a genome-wide test using fixed cutoffs), one can fine-tune the cutoffs utilizing this strategy.

In the independent test, half genes have p-values less than 0.05, and most of them have p-values >0.01. Purely based on the number of genes and their p-values, the validation rate is relatively low, and the usability of the biomarkers is questionable. The p-values are affected by the sample size, which should be described in the main text. As I mentioned above, the authors should look for more data to verify these biomarkers.

How univariable and multivariable tests work should be clearly described in the main text. In a multivariable test, which variables were used?

Here is another suggestion. To maximize the use of continuous expression values, the author can build a multivariable model (e.g., logistic regression) integrating multiple genes.

Minor comments:
The background should be revised extensively. For example, the description of the linear model should be removed. X1...Xn should be removed. More introduction to how other researchers used TCGA to find expression biomarkers should be presented. Detailed introduction of TCGA-related tools should be removed or moved to Methods.

In line 154 paragraph, what does unclean data mean, missing values or zero expression? The three challenges are not “challenges” at all. The authors exaggerated the difficulties here.
In Figure 1, the diagram should have direction. The file names are not interesting. RNAseq (20500 genes) should be gene expression values (20500 genes)

In Figure 2, unlabeled scales on the x-axis.